# The OpenNeuro resource for sharing of neuroscience data

Christopher J Markiewicz[1], Krzysztof J Gorgolewski[1], Franklin Feingold[1], Ross Blair[1], Yaroslav O Halchenko[2], Eric Miller[3], Nell Hardcastle[3], Joe Wexler[1], Oscar Esteban[1,4], Mathias Goncavles[1], Anita Jwa[1], Russell Poldrack[1]*

[1]Department of Psychology, Stanford University, Stanford, United States; [2]Department of Psychological & Brain Sciences, Dartmouth College, Hanover, United States; [3]Squishymedia, Portland, United States; [4]Lausanne University Hospital and University of Lausanne, Lausanne, Switzerland

**Abstract** The sharing of research data is essential to ensure reproducibility and maximize the impact of public investments in scientific research. Here, we describe OpenNeuro, a BRAIN Initiative data archive that provides the ability to openly share data from a broad range of brain imaging data types following the FAIR principles for data sharing. We highlight the importance of the Brain Imaging Data Structure standard for enabling effective curation, sharing, and reuse of data. The archive presently shares more than 600 datasets including data from more than 20,000 participants, comprising multiple species and measurement modalities and a broad range of phenotypes. The impact of the shared data is evident in a growing number of published reuses, currently totalling more than 150 publications. We conclude by describing plans for future development and integration with other ongoing open science efforts.

*For correspondence:
russpold@stanford.edu

## Introduction

There is growing recognition of the importance of data sharing for scientific progress (***National Academies of Sciences, Engineering, and Medicine, Policy and Global Affairs, Board on Research Data and Information, Committee on Toward an Open Science Enterprise, 2018***). However, not all shared data are equally useful. The FAIR principles (***Wilkinson et al., 2016***) have formalized the notion that in order for shared data to be maximally useful, they need to be findable, accessible, interoperable, and reusable. An essential necessity for achieving these goals is that the data and associated metadata follow a common standard for organization, so that data users can easily understand and reuse the shared data. Here, we describe the OpenNeuro data archive (RRID:SCR_005031), accessible at https://openneuro.org, which enables FAIR-compliant data sharing for a growing range of neuroscience data types (currently including magnetic resonance imaging [MRI], electroencephalography [EEG], magnetoencephalography [MEG], and positron emission tomography [PET]) through the use of a common community standard, the Brain Imaging Data Structure (BIDS) (RRID:SCR_016124; ***Gorgolewski et al., 2016***).

Starting with early pioneering efforts by Gazzaniga and Van Horn to establish an fMRI Data Center in 1999 (***Van Horn and Gazzaniga, 2013***), data sharing has become well established in the domain of neuroimaging (***Milham et al., 2018***; ***Poldrack and Gorgolewski, 2014***; ***Poline et al., 2012***). A major impetus for the growth of data sharing was the International Neuroimaging Data Sharing Initiative (***Mennes et al., 2013***), which published a landmark paper in 2010 (***Biswal et al., 2010***) demonstrating the scientific utility of a large shared resting fMRI dataset. The most prominent recent examples have been large-scale prospective data sharing projects, including the Human Connectome Project (HCP) (***Van Essen et al., 2013***), the NKI-Rockland sample (***Nooner et al., 2012***), Adolescent Brain Cognitive

Development study (*Casey et al., 2018*), and the UK Biobank (*Littlejohns et al., 2020*). These datasets have provided immense value to the field and have strongly demonstrated the utility of shared data. However, their scientific scope is necessarily limited, given that each dataset includes only a limited number of imaging tasks and measurement types. Beyond these large focused data sharing projects, there is a 'long tail' of smaller neuroimaging datasets that have been collected in service of specific research questions. Making these available is essential to ensure reproducibility as well as to allow aggregation across many different types of measurements in service of novel scientific questions. The OpenNeuro archive addresses this challenge by providing researchers with the ability to easily share a broad range of neuroimaging data types in a way that adheres to the FAIR principles.

## Goals and principles

The OpenNeuro archive evolved from the OpenfMRI archive (*Poldrack et al., 2013*), which was focused solely on the sharing of task-based human fMRI data. Some of the principles behind Open-Neuro were inherited from OpenfMRI, whereas others grew out of our experiences in that project as well as from new developments in the domain of open science.

## Minimal restrictions on sharing

There is a range of restrictiveness across data archives with regard to their data use agreements (*Jwa and Poldrack, 2021*). At one end of the spectrum are highly restricted databases such as the Alzheimer's Disease Neuroimaging Initiative, which requires researchers to submit their scientific question for review and requires the consortium to be included as a corporate author on any publications. OpenNeuro represents the other pole of restrictiveness, by releasing data (by default) under a Creative Commons Zero (CC0) Public Domain Dedication which places no restrictions on who can use the data or what can be done with them. While not legally required, researchers using the data are expected to abide by community norms and cite the data following the guidelines included within each dataset. The primary motivation for this policy is that it makes the data maximally accessible to the largest possible number of researchers and citizen-scientists.

## Standards-focused data sharing

To ensure the utility of shared data for the purposes of efficient discovery, reuse, and reproducibility, standards are required for data and metadata organization. These standards make the structure of the data clear to users and thus reduce the need for support by data owners and curation by repository owners, as well as enabling automated QA, preprocessing, and analytics. Unfortunately, most prior data sharing projects have relied upon custom organizational schemes, which can lead to misunderstanding and can also require substantial reorganization to adapt to common analysis workflows. The need for a clearly defined standard for neuroimaging data emerged from our experiences in the OpenfMRI project; while the repository had developed a custom scheme for data organization and file naming, this scheme was ad hoc and limited in its coverage, and datasets often required substantial manual curation (involving laborious interaction with data owners). In addition, there was no way to directly validate whether a particular dataset met the standard.

For these reasons, we focused at the outset of the OpenNeuro project on developing a more robust data organization standard that could be implemented in an automated validator. We engaged a large group of researchers from the neuroimaging community to establish a standard that ultimately became the BIDS (*Gorgolewski et al., 2016*), which is now a highly successful community standard for a broad and growing range of neuroimaging data types. BIDS defines a set of schemas for file and folder organization and naming, along with a schema for metadata organization. The framework was inspired by the existing data organization frameworks used in many research laboratories, so that transitioning to the standard is relatively easy for most researchers. One of the important features of BIDS is its extensibility; using a scheme inspired by open-source software projects, community members can propose extensions to BIDS that encompass new data types. To date, modality extensions include MEG (*Niso et al., 2018*), scalp EEG (*Pernet et al., 2019*), intracranial EEG (*Holdgraf et al., 2019*), PET (*Norgaard et al., 2021*), and arterial spin labeling MRI. In addition to standards for raw data, the BIDS community has also developed a standard for the organization of the outputs of processing operations (known as 'BIDS Derivatives'), providing a framework for sharing processed as well as raw data.

While BIDS and OpenNeuro are now independent projects, there is a strongly synergistic relationship. All data uploaded to OpenNeuro must first pass a BIDS validation step, such that all data in OpenNeuro are compliant with the BIDS specifications at upload time. Conversely, the OpenNeuro team has made substantial contributions to the BIDS standard and validator. The BIDS standard has been remarkably successful, with tens of thousands of datasets now available in the format, including but not limited to those contained in the OpenNeuro database. As a consequence, this model maximizes compatibility with processing and analysis tools (*Gorgolewski et al., 2017*), but more importantly, it effectively minimizes the potential for data misinterpretation (e.g., when owner and reuser have slightly different definitions of a critical acquisition parameter). Through the adoption of BIDS, OpenNeuro has moved away from project- or database-specific data structures designed by the owner or the distributor (as used in earlier projects such as OpenfMRI and HCP) and toward a uniform and unambiguous representation model agreed upon by the research community prior to sharing and reuse.

## FAIR sharing

The FAIR principles (*Wilkinson et al., 2016*) have provided an important framework to guide the development and assessment of open data resources. OpenNeuro implements each of these principles.

Findable: Each dataset within OpenNeuro is associated with metadata, both directly from the BIDS dataset along with additional dataset-level metadata provided by the submitter at time of submission. Both data and metadata are assigned a persistent unique identifier (Digital Object Identifier [DOI]). Within the repository, a machine-readable summary of BIDS metadata is collected by the BIDS validator and indexed with an ElasticSearch mapping. In addition, dataset-level metadata are exposed according to the schema.org standard, which allows indexing by external resources such as Google Dataset Search.

Accessible: Data and metadata can be retrieved using a number of access methods (directly from Amazon S3, using the OpenNeuro command line tool, or using DataLad) via standard protocols (http/https). Metadata are also accessible programmatically via a web API. Metadata remain available even in the case that data must be removed (e.g., in cases of human subjects concerns). No authentication is necessary to access the data.

Interoperable: The data and metadata use the BIDS standard to ensure accessible representation and interoperation with analysis workflows, such as BIDS Apps (*Gorgolewski et al., 2017*). Ongoing work is extending the metadata representation to use richer formats and to link to relevant FAIR ontologies or vocabularies.

Reusable: The data are released with a clear data use agreement (currently defaulting to a CC0 public domain dedication). Through use of the BIDS standard, the data and metadata are consistent with community standards in the field.

## Data versioning and preservation

OpenNeuro keeps track of all changes in stored datasets and allows researchers to unambiguously report the exact version of the data used for any analysis. OpenNeuro preserves all versions of the data through the creation of 'snapshots' that unequivocally point to one specific point in the lifetime of a dataset. Data management and snapshots are supported by DataLad (RRID:SCR_003931; *Halchenko et al., 2021*), a free and open-source distributed data management system (*Hanke et al., 2021*).

## Protecting privacy and confidentiality of data

There is a direct relationship in data sharing between the openness of the data and their reuse potential; all else being equal, data that are more easily or openly available will be more easily and readily reused. However, all else is not equal, as openness raises concern regarding risks to subject privacy and confidentiality of data in human subjects research. Researchers are ethically bound to both minimize the risks to their research participants (including risks to confidentiality) and to maximize the benefits of their participation (United States. National Commission for the *States, 1978*). Because sharing of data will necessarily increase the potential utility of the data, researchers are ethically bound to share human subject data unless the benefits of sharing are outweighed by risks to the participant (*Brakewood and Poldrack, 2013*).

In general, risks to data privacy and confidentiality are addressed through deidentification of the data to be shared. For example, under the Health Insurance Portability and Accountability Act of 1996 (HIPAA) in the United States, deidentification can be achieved through the removal of any of 18 personal identifiers, unless the researcher has knowledge that the remaining data could be re-identified (known as the 'safe harbor' method). With regard to neuroimaging data, a particularly challenging feature is the facial structure that is present in some forms of imaging data, such as structural MRI images. It is often possible to reconstruct facial structures from these images, and there are proofs of concept that such data could be used to re-identify individuals from photographic databases (*Schwarz et al., 2019*). It is thus essential to remove any image features that could be used to reconstruct facial structure (*Bischoff-Grethe et al., 2007*). For this reason, all MRI data shared through OpenNeuro must have facial features removed prior to upload, in addition to the 18 personal identifiers outlined by HIPAA. An exception is provided in cases where an investigator has explicit permission to openly share the data without defacing, usually when the data are collected by the investigator themself. At present, data are examined by a human curator to ensure that this requirement has been met. In the future, we plan to deploy an automated face detection tool (*Bansal et al., 2020*) to detect any uploads that inadvertently contain facial features.

Truly informed consent requires that subjects be made aware that their data may be shared publicly, and that confidentiality cannot be absolutely guaranteed in the future. For this reason, we recommend that researchers planning to share their data via OpenNeuro use a consent form based on the Open Brain Consent (*Bannier et al., 2021*), which includes language that ensures subject awareness of the intent to share and its potential impact on the risk of participating. Of note, the Open Brain Consent has recently been adapted to include a data usage agreement that accommodates the European Union's General Data Protection Regulation (GDPR 2016/679); however, data collected in countries covered by GDPR cannot be shared through OpenNeuro at present due to the requirement for restrictive data use agreements that are not currently supported by OpenNeuro.

## Open source

The entirety of the code for OpenNeuro is available under a permissive open-source software license (MIT License) at https://github.com/OpenNeuroOrg/openneuro. This enables any researcher who wishes to reuse part or all of the code or to run their own instance of the platform.

## Data submission and access

*Figure 1* outlines the steps required for sharing a dataset using OpenNeuro. Once shared, data can be accessed by several available mechanisms.

Web download: Each snapshot is associated with a link that provides immediate downloading of the dataset.

DataLad. DataLad (*Halchenko et al., 2016*) is a decentralized data management system built on top of git and git-annex. Through DataLad, researchers may install a complete copy of a dataset, while deferring the retrieval of file contents until needed, permitting lightweight views of large datasets. OpenNeuro's versioned snapshots are implemented as git tags, which allows specific versions to be easily retrieved or compared. The decentralized protocol also allows mirrors of the datasets to be hosted on GitHub and https://datasets.datalad.org, ensuring access during service interruptions of the OpenNeuro website.

OpenNeuro command line tool: The OpenNeuro command line tool provides access to the latest snapshot of all datasets, and is generally more stable than browser downloads for large datasets.

Amazon S3: The latest snapshot as well as all previous versions of a dataset may be fetched using the Amazon Web Services (AWS) clients or directly via https.

## User support

Support for individual datasets: Data users sometimes have questions regarding particular datasets. In order to facilitate discussion of these issues and to make those discussions available to the entire community, a discussion forum is provided on each dataset page. The dataset owner is automatically notified by email of any questions that are posted. In addition, users can 'follow' a dataset of interest and receive notifications of any comments posted to the dataset.

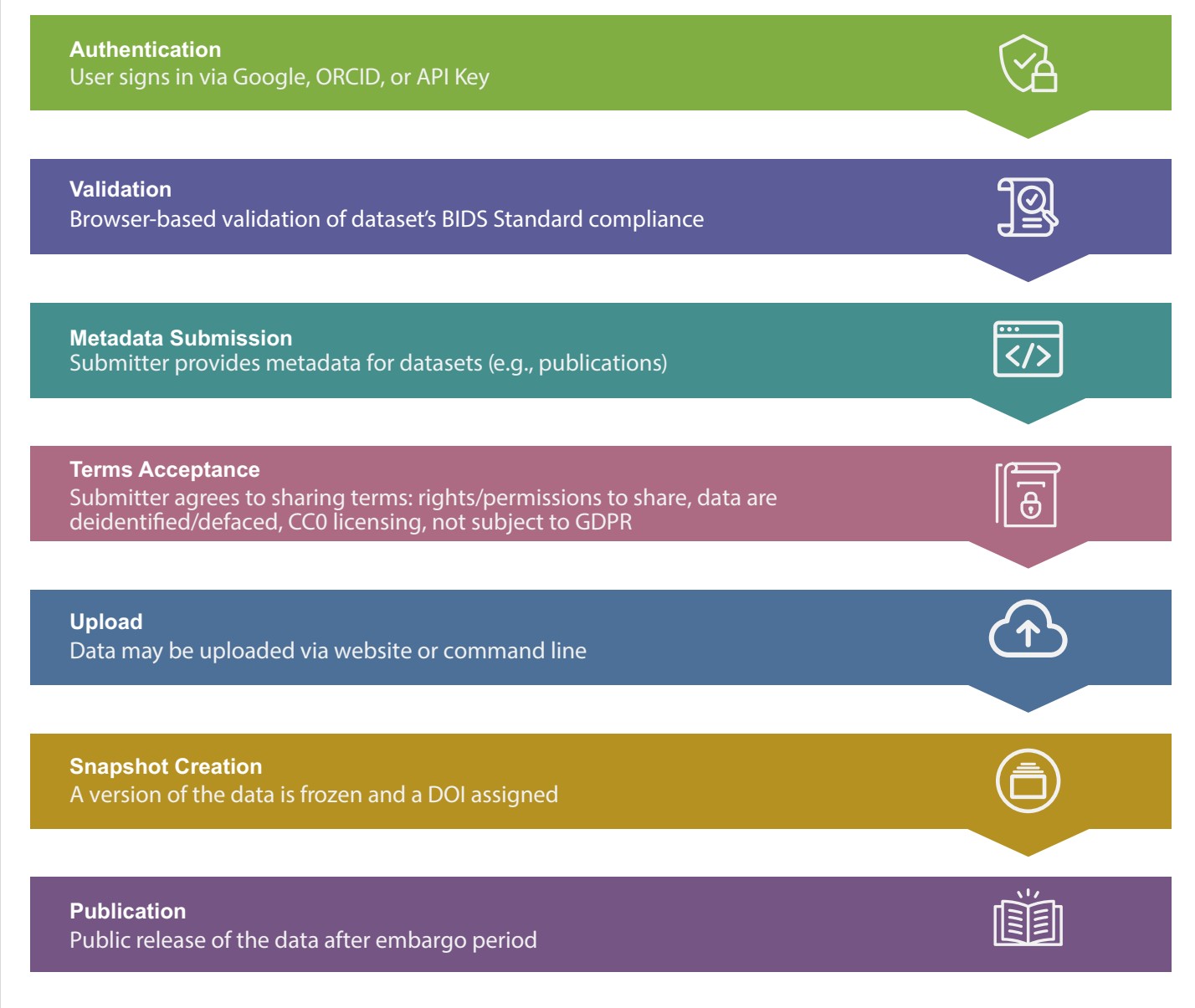

**Authentication**
User signs in via Google, ORCID, or API Key

**Validation**
Browser-based validation of dataset's BIDS Standard compliance

**Metadata Submission**
Submitter provides metadata for datasets (e.g., publications)

**Terms Acceptance**
Submitter agrees to sharing terms: rights/permissions to share, data are deidentified/defaced, CC0 licensing, not subject to GDPR

**Upload**
Data may be uploaded via website or command line

**Snapshot Creation**
A version of the data is frozen and a DOI assigned

**Publication**
Public release of the data after embargo period

**Figure 1.** A schematic overview of the data upload process.

The online version of this article includes the following figure supplement(s) for figure 1:

**Figure supplement 1.** Word clouds based on Cognitive Atlas terms for psychological concepts (top) and tasks (bottom) identified from titles and README files associated with OpenNeuro datasets.

**Figure supplement 2.** Word clouds based on Cognitive Atlas terms for psychological concepts (top) and tasks (bottom) identified from titles and README files associated with OpenNeuro datasets.

Site support: Two mechanisms are provided for users of the OpenNeuro site to obtain help with site issues. First, a helpdesk is available directly from the site, through which users can submit specific help questions. Second, users are recommended to post general questions to the Neurostars.org question and answer forum, so that the answers will be available to the entire community.

## Data processing

Data processing was initially envisioned as an incentive for researchers to share their data, and the OpenNeuro site was launched in 2017 with the ability to perform cloud-based data processing using a limited set of analysis workflows. This feature was disabled in 2018, after an overhaul of the site's initial

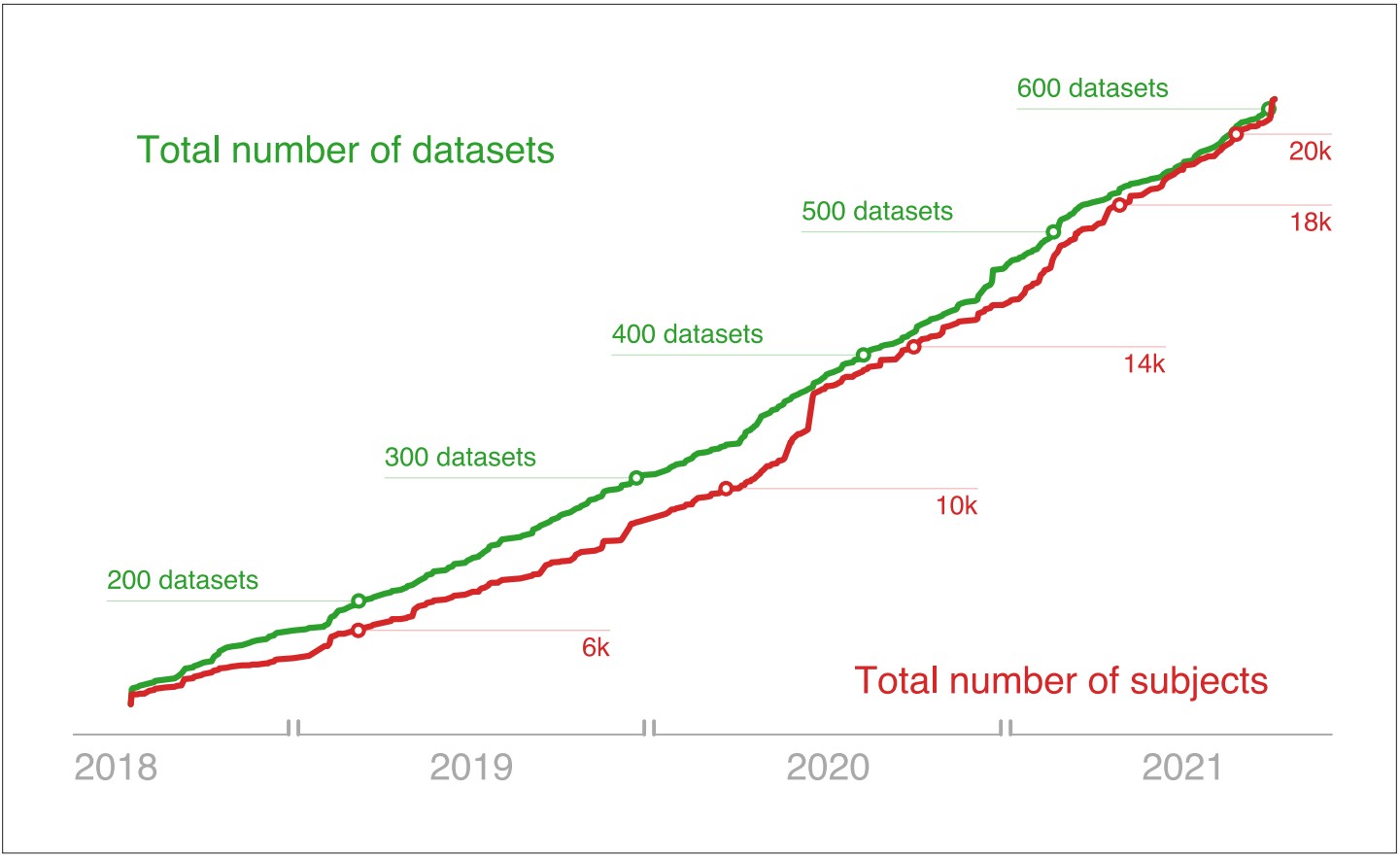

**Figure 2.** The volume of data available on OpenNeuro has shown a steady growth since its opening started operations in 2017. Shown are figures from July 2018, when all data were migrated to a new DataLad storage backend, through the present date. The green line illustrates the cumulative growth in total number of datasets, and the red line shows the aggregate of subjects (in thousands).

storage infrastructure. At that time, we determined that it would be preferable to collaborate with an existing platform dedicated to cloud processing rather than rebuilding our own execution platform. At present, OpenNeuro has partnered with the Brainlife.io platform (RRID:SCR_020940), which provides a large set of cloud-based neuroimaging workflows for data analysis and visualization. Data hosted on OpenNeuro can be easily imported into Brainlife for analysis, and more than 400 OpenNeuro datasets are cached for quick access; in the first 6 months of 2021, more than 700 analyses were performed on these datasets. In the future we plan to partner with additional platforms, including the NEMAR platform for EEG/MEG analysis; the availability of the data via DataLad and Amazon S3 also enables any platform to make the data available to their users without requiring any agreement or effort from OpenNeuro.

## Results

### Usage and impact

The OpenNeuro site was launched in June 2017, and was originally seeded with all of the datasets previously shared through OpenfMRI, after converting them to the BIDS standard. All data presented below are current as of October

**Table 1.** Number of datasets by imaging modality; additional modalities present in fewer than three datasets are not included here.

| Modality | Number of datasets |
|---|---|
| Anatomical MRI | 501 |
| Functional MRI | 445 |
| Electroencephalography | 81 |
| Diffusion-weighted MRI | 53 |
| Magnetoencephalography | 23 |
| Positron emission tomography | 10 |
| Intracranial EEG | 8 |
| Arterial spin labeling MRI | 3 |

9, 2021. The database contains 604 datasets comprising data from 20,989 individual participants. *Figure 2* shows cumulative figures for numbers of datasets and subjects since 2018, demonstrating sustained and continual growth in the archive since its inception.

The overwhelming majority of datasets are from humans (574 datasets, 95%), with a small but growing number of nonhuman species including mouse (17 datasets), rat (6 datasets), nonhuman primates (2 datasets), dogs (1 dataset), and juvenile pigs (1 dataset). *Table 1* presents data for the prevalence of different modalities; while the majority of datasets include some form of MRI data, other supported modalities are present including electrophysiological measures and PET.

OpenNeuro is a recommended data repository for a number of publishers and journals, including Nature Scientific Data, PLOS, eLife, F1000 Research, Gigascience, BioMed Central, American Heart Association, and Wellcome Open Research. The database contains 407 DOIs for publications associated with datasets (including both primary scientific publications and data descriptors).

## Multiple dimensions of 'big data'

Discussions of 'big data' in neuroimaging (*Poldrack and Gorgolewski, 2014*; *Smith and Nichols, 2018*) have largely focused on datasets including large numbers of individuals. While these analyses are essential for robust population inference, it is also important to recognize that large numbers of subjects are only one dimension over which a neuroimaging dataset can be 'big'. Here, we will define the number of subjects as the 'width' of the dataset, the number of different phenotypes measured for each individual as the 'breadth' of the dataset, and the number of measurements per individual as the 'depth' of the dataset.

The OpenNeuro database is distinguished by sharing datasets that are extensive along each of these dimensions (see *Figure 3*). With regard to width, the median dataset size is 23 subjects, with 31 studies having sample sizes larger than 100, and a maximum sample size of 928. With regard to breadth, notable datasets include the BOLD5000 dataset (*Chang et al., 2019*), which includes data from subjects viewing a total of 5000 natural images; the Individual Brain Charting dataset (*Pinho et al., 2020*; *Pinho et al., 2018*), which includes data from individuals each completing 24 different tasks, and the Multidomain Task Battery dataset (*King et al., 2019*), which includes data from individuals each completing 26 tasks. With regard to depth, the database currently includes the MyConnectome dataset (*Poldrack et al., 2015*), which includes extensive task, resting, and diffusion MRI data from more than 100 sessions for a single individual; the Midnight Scan Club dataset (*Gordon et al., 2017*), which includes extensive task and resting fMRI data from 10 individuals; and a number of other dense scanning datasets (*Gonzalez-Castillo et al., 2015*; *Newbold et al., 2020*; *Salehi et al., 2020*).

Another unique feature of OpenNeuro is the breadth of phenotypes across datasets. To further characterize this, we searched the text associated with OpenNeuro datasets to identify terms related to psychological concepts and tasks as defined in the Cognitive Atlas ontology (*Poldrack et al., 2011*). Word clouds showing the top terms identified in this analysis are shown in *Figure 1—figure supplement 2*. This analysis shows a broad range of tasks and concepts associated with these datasets, highlighting the substantial conceptual and methodological breadth of the archive.

## Data reuse

OpenNeuro has distributed a substantial amount of data; from May 2020 through April 2021, a total of 406 terabytes of data were distributed. Because data reuse is not directly measurable, we utilize published reuse of the shared data as a proxy. To identify published reuses of OpenNeuro data, we used Google Scholar and CrossRef to identify potential reuses, and then manually examined them to confirm that they were a legitimate reuse (as opposed to a primary publication of the data or data descriptor); note that this is an underestimate since many papers during this period reported analyses of data downloaded from OpenfMRI, which would not have been identified in our searches. We identified 165 publications that reused OpenNeuro datasets; this showed a sharp increase over time (see *Figure 4*). Of these publications, 112 were journal or conference papers, 42 were preprints, and 11 were other types of publications (such as theses or project reports). A total of 111 OpenNeuro datasets were reused at least once, with the most popular dataset (*Poldrack et al., 2016*) appearing in 28 published reuses. A significant number of publications reused multiple datasets; 31 of the 165 papers reused at least two datasets, with a maximum of 40 datasets reused (*Esteban et al., 2019*). Collecting these data from scratch would have required more than 21,000 individual subject visits; at

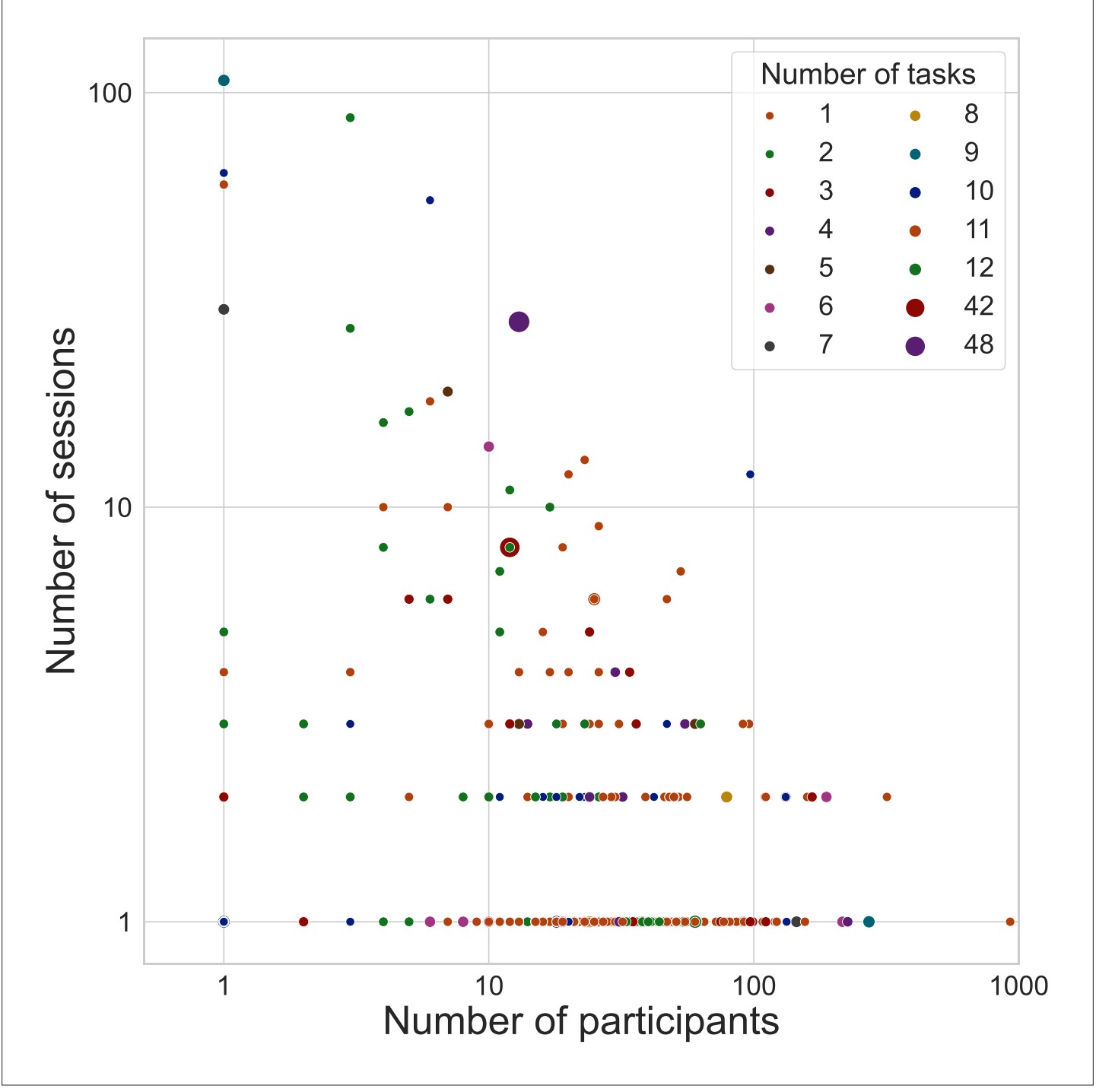

**Figure 3.** OpenNeuro datasets vary substantially in number of participants (X axis), number of sessions per participant (Y axis), and number of tasks per participant (size/color of datapoints); axes are log-scaled for easier visualization. Results are based on metadata derived directly from the 502 OpenNeuro datasets available via DataLad as of 10/9/2021.

an estimated scanning cost of $1000/session (based on the conservative cost estimate from *Milham et al., 2018*), this represents a total data reuse value of nearly 21 million US dollars. These reuses have a total of 1329 citations (according to Google Scholar as of June 15, 2021); the most highly cited reuse (*Esteban et al., 2019*) has more than 500 citations.

The published reuses of OpenNeuro data span from basic neuroscience to methodological studies and software development. In particular, several studies demonstrate how OpenNeuro data have

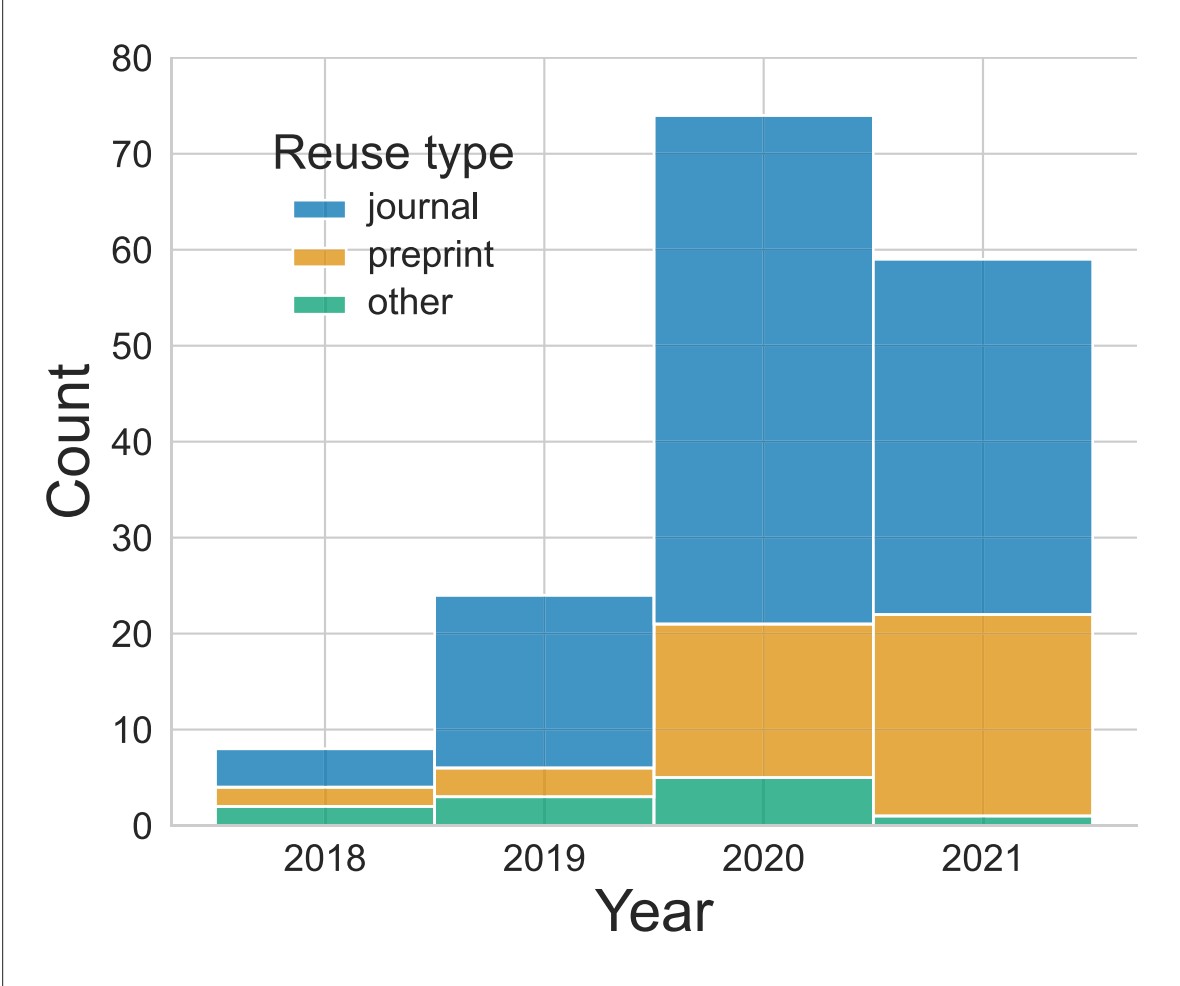

**Figure 4.** Published reuses of OpenNeuro datasets, split by the type of reuse. Note that the final bar includes only reuses identified through June 2021.

enabled new insights into brain function. For example, *Martins et al., 2021*, used structural MRI data from several OpenNeuro datasets along with other shared data to examine different patient groups suffering from physical pain or depression. Their analyses demonstrated a specific pattern of anatomical change common to patients with pain syndromes but distinct from depression. This kind of analysis highlights the way in which OpenNeuro enables researchers to combine smaller datasets in order to test hypotheses using convergent data, which can help overcome the confounds and biases present in any particular study as well as increasing statistical power. Other basic neuroscience studies have used OpenNeuro data to model the role of temporal context in forgetting (*Chien and Honey, 2020*), characterize the role of edge communities in brain networks (*Faskowitz et al., 2020*), understand the relationship between functional connectivity and sustained attention (*Rosenberg et al., 2020*), and to demonstrate that functional parcellation changes as a function of task (*Salehi et al., 2020*).

Data from OpenNeuro have been particularly useful for the development of new software tools. *Esteban et al., 2019*, used the breadth and variety of datasets in the archive to assess the robustness of the fMRIPrep preprocessing workflow to many different fMRI datasets, incorporating a total of 40 datasets from OpenNeuro. Importantly, these datasets were used in an iterative manner to improve the robustness of the tool; thus, the breadth of the data were essential both for assessment and for improvement of the tool. Without OpenNeuro (and BIDS), amassing such a large and diverse group of datasets would have required immense efforts to reach out to many different research groups, request their data, and then format the data for common usage, whereas with OpenNeuro the entirety of these datasets can be downloaded automatically within a number of hours, immediately ready for analysis. Other software development projects have taken advantage of some of the particular unique datasets

in OpenNeuro; for example, *Takeda et al., 2019*, took advantage of a unique dataset that combines EEG, MEG, and MRI data on the same individuals (*Wakeman and Henson, 2015*) to demonstrate the broad range of functions of their VBMEG toolbox. Other software publications using OpenNeuro data include FastSurfer (*Henschel et al., 2020*) for structural MRI analysis, and Brainstorm (*Tadel et al., 2019*) for MEG/EEG analysis.

The data in OpenNeuro have been particularly useful for methodological researchers. One prominent example was published by *Bowring et al., 2019*, who examined how the use of different analysis software impacted statistical results from fMRI activation analyses. Their study included an in-depth analysis of the publications associated with each of 55 datasets, in order to identify studies with analysis pipelines and activation results that could be easily compared with their multi-platform results. Based on this process, they selected three datasets and processed each using several different analysis pipelines; their results highlighted substantial similarity in unthresholded maps but substantial discordance in thresholded maps, highlighting the need for better understanding of the impact of software packages on statistical results. Another example that would have been challenging to perform without OpenNeuro was published by *Dadi et al., 2020*, who developed a set of functional atlases using 27 datasets. This breadth allowed them to ensure that the specific features of the atlas were not driven by any particular dataset or task. Other examples include studies that used OpenNeuro data to assess the impact of confound regression on fMRI signals and develop new methods for confound modeling (*Aquino et al., 2020*), and to develop and benchmark new methods for multiple comparison correction (*Spisák et al., 2019*).

## Discussion

The OpenNeuro data archive plays an important role in advancing neuroscience research and ensuring its reproducibility by enabling the sharing of a broad range of neuroscience data types according to the FAIR principles. Its tight integration with the community-driven BIDS standard enhances the ease of sharing, the reusability of the shared data, and the extensibility of the archive in the future. The shared data have enabled a growing number of publications that provide novel neuroscientific insights, as well as supporting novel methodological advances and software development.

### Lessons learned

The experiences of our group in developing the OpenNeuro project have provided a number of lessons that may be useful more generally for researchers interested in establishing a culture of data sharing within their scientific subdomain.

Foremost, we have found that the use of a common community-driven format for data organization is essential to effective sharing. In our case, the BIDS standard has enabled data owners to easily share a growing range of data types (through the use of client-side validator), and has enabled researchers to easily reuse the data. Because any dataset that passes the validator can be shared, the community's efforts on extending the standard (which are implemented in the validator) has provided a steady stream of additions to the types of data that OpenNeuro can share. Another important point is that data sharing does not only include sharing with other researchers, but also with one's own research group in the future; thus, the use of a well-structured data standard can help researchers ensure that data collected by current lab members can be effectively utilized by other lab members in future, as well as making it easy to share the data beyond one's own lab. On the flipside, we continue to see that conversion of data into the BIDS standard remains a stumbling block for many researchers; the continued development of conversion tools is necessary to support these researchers.

Second, we have found that 'it takes an ecosystem' to make data sharing successful. OpenNeuro is only one of the data sharing projects within the field of neuroimaging, and each of the projects has its own particular features and advantages, but together these projects have increasingly led the field to view data sharing as a net positive for our field. In addition, the availability of these data resources has allowed others to build projects that support new mechanisms for data representation and distribution (such as the DataLad project) and new platforms for analysis (such as Brainlife.io). Together, these tools have provided researchers with additional incentives to share their data via OpenNeuro through its deep integration with those projects. While we believe that sharing is most effective when it is most open, we also realize that some researchers will be unable to share their data on OpenNeuro for

ethical or regulatory reasons; for this reason, we believe that a variety of data sharing resources that vary in their sharing policies (*Jwa and Poldrack, 2021*) will remain essential to support the broadest possible degree of data sharing.

Finally, we would highlight the importance of domain-specific data repositories that support a particular research community. All of the sharing activities accomplished using OpenNeuro could in principle have been accomplished using more general data sharing repositories (such as Figshare or Dryad). A unique benefit of OpenNeuro has been in making a large number of datasets easily findable by researchers, rather than requiring a trawl through a much larger body of datasets to find ones that are relevant. By developing upload and download systems that are tailored for imaging data, OpenNeuro has also greatly lowered the barrier to sharing and reusing data. These benefits argue for the continued need for domain-specific data sharing projects designed in close consultation with researchers in the area. Domain specificity has also allowed OpenNeuro to nurture a community around the resource. Through our social media presence, we have engaged the community with regular blog posts that highlight the most open and sharing labs over the previous 6 months to promote more social incentives to sharing.

## Long-term sustainability

A continual challenge for any investigator-initiated data repository is the long-term sustainability of the archive, in order to ensure researchers' trust in the platform (*Lin et al., 2020*). The ongoing costs of running a repository are substantial, primarily due to the continuing cost of technological upkeep of a web platform with regard to security and stability, as well as the ongoing costs of storage and bandwidth on cloud platforms or hardware maintenance when using on-premise computing systems. Performant web applications require the use of cutting-edge software tools, which can often become deprecated or unstable over time, leading to substantial technical debt that must be continually addressed to maintain stable and secure operation.

One major challenge for repositories that are reliant upon federal grants is the usual 3-year funding period, in addition to the preference of standard grant mechanisms for funding novel projects rather than ongoing maintenance and operations. One welcome development has been the instigation of longer-term funding for data archives through the US BRAIN Initiative (*Koroshetz et al., 2018*), which has explicitly dedicated funding to the development and long-term sustainability of data archives for neuroscience data. These renewable 5-year grants (of which OpenNeuro is one of the recipients) provide a much-needed longer-term funding source for data repositories.

Another resource for longer-term sustainability is institutional data repositories, which are increasingly available at many universities. OpenNeuro is working with the Stanford Digital Repository to develop a plan to deposit all raw datasets within the university's archive, which would provide a digital backstop to the archive's cloud storage.

OpenNeuro has also been fortunate to be part of the Amazon Public Datasets project (https://registry.opendata.aws/openneuro/), which has provided free data storage and bandwidth for the openly available datasets in the OpenNeuro archive.

## Current limitations and future directions

There are a number of additional features planned for future development. These include:

Enhanced metadata: At present, a limited amount of dataset-level metadata is collected beyond that present within the BIDS metadata. Working with the CEDAR Metadata Center (*Musen et al., 2015*), we plan to add the ability for researchers to enter additional metadata that is linked to standard ontologies, including those being developed for BIDS data in the context of the Neuroimaging Data Model (*Maumet et al., 2016*). These annotations will provide the basis for more powerful queries of the archive.

Sharing of derivatives: At present, OpenNeuro only shares raw data. However, the availability of a BIDS standard for the outputs of data processing (i.e. 'derivative' data) now provides the ability to include derivative data within a BIDS dataset. We plan to enable researchers to share derivatives, for example, allowing the sharing of preprocessed MRI data in addition to raw data. This will greatly enhance the reuse of data by researchers who do not have the resources or expertise to preprocess these complex datasets as well as provide a standard baseline for downstream analyses, reducing the potential effects of analytic flexibility (*Botvinik-Nezer et al., 2020*; *Bowring et al., 2019*).

Bringing computing to data: The availability of the OpenNeuro data on the AWS allows researchers direct access to computing on the data, but doing so requires a substantial degree of cloud computing expertise. To ease the application of computing to the data, we plan to adapt the DANDI Hub infrastructure developed by the Distributed Archives for Neurophysiology Data Integration (DANDI: https://www.dandiarchive.org/), which will allow direct access to the data via a Jupyter notebook.

Beyond MRI data: Driven by the initial seeding of data from OpenfMRI, and reflecting the fact that BIDS was originally MRI-centric, the data currently available from OpenNeuro are heavily skewed toward MRI, and fMRI in particular (*Table 1*). However, BIDS is quickly expanding to other modalities that can readily be uploaded to OpenNeuro, and there has been a rapid increase in sharing of other modalities; for example, more than 60 EEG datasets have been deposited since the publication of the BIDS-EEG standard in 2019 (*Pernet et al., 2019*). This organic expansion beyond MRI will be supported with the necessary adaptations (e.g., online visualization of new modalities) of OpenNeuro's user interface.

## Conclusion

Data sharing ensures the transparency and reproducibility of scientific research, and allows aggregation across datasets that improves statistical power and enables new research questions. The OpenNeuro repository plays a central role in the data sharing ecosystem by promoting maximally open sharing of data, and by enhancing open availability of data from a wide range of datasets spanning. The growth and impact of the repository demonstrate the viability of minimally restrictive sharing, and the importance of common standards such as BIDS for the effective sharing and reuse of data.

# Materials and methods
## OpenNeuro infrastructure

Code for the OpenNeuro platform is available at (https://github.com/OpenNeuroOrg/openneuro). The application utilizes a cloud-based containerized architecture and is built in JavaScript and Python with a MongoDB database for application data storage. OpenNeuro is hosted on AWS using the Kubernetes container orchestration platform. Services are deployed as containers and integrated via a JavaScript GraphQL API gateway and the AWS Application Load Balancer. Several clients access this API, the React website, OpenNeuro command line interface, and an ElasticSearch indexer. Datasets are stored as DataLad repositories and managed by a Python backend service container. Each DataLad repository is assigned to a ZFS pool backed by AWS Elastic Block Store. This allows DataLad versioning and filesystem level access to datasets with existing processing and validation tools. Persistent metadata such as user accounts and permissions are maintained in a MongoDB database. Ephemeral caching is provided by Redis. Search indexes, performance monitoring, and logging are implemented with ElasticSearch. CloudFront is used as a global cache and network to provide global presence.

## Content analysis

Data regarding OpenNeuro contents and usage were current as of October 9, 2021. Code and data needed to execute all analyses and generate all figures are available from https://doi.org/105281/zenodo5559041.

Reuse analyses: Potential reuses were identified by first searching Google Scholar for the term "OpenNeuro"; note that this will exclude any paper that mention 'OpenfMRI' instead of OpenNeuro, thus the reported results are underestimates of the true impact of the data, given that many of the datasets in OpenNeuro came from OpenfMRI. Papers matching this search were examined manually to confirm that they had reused data; data descriptor papers were excluded from further analysis. Citation counts were obtained from Google Scholar using the Python package 'scholarly'.

Dataset size analyses: Dataset size analyses were performed using DataLad to obtain the full BIDS metadata for the 502 datasets available as of October 9, 2021, and then using PyBIDS (*Yarkoni et al., 2019*) to load the metadata for each dataset.

# Acknowledgements

The work described here has been supported by the National Institute of Mental Health of the National Institutes of Health under Award Numbers R24MH117179 and R24MH114705. The content is solely the responsibility of the authors and does not necessarily represent the official views of the National Institutes of Health. Development of OpenNeuro and OpenfMRI was also supported by a grant from the Laura and John Arnold Foundation, and the National Science Foundation (OAC-1131441). Sharing of OpenNeuro datasets has been enabled by support from AWS. We would like to thank all of the users who have uploaded data to OpenNeuro. Thanks to Franco Pestili for providing usage data on Brainlife.io, and Nico Dosenbach, Michael Hawrylycz, Karel Svoboda, and Armin Thomas for helpful comments on an earlier draft.

## Additional information

### Competing interests

Eric Miller: is owner of Squishymedia which is funded to perform software development work on OpenNeuro.. Nell Hardcastle: is an employee of Squishymedia which is funded to perform software development work on OpenNeuro.. The other authors declare that no competing interests exist.

### Funding

| Funder | Grant reference number | Author |
| --- | --- | --- |
| National Institute of Mental Health | R24MH117179 | Russell Poldrack |
| National Institute of Mental Health | R24MH114705 | Russell Poldrack |

The funders had no role in study design, data collection and interpretation, or the decision to submit the work for publication.

### Author contributions

Christopher J Markiewicz, Conceptualization, Project administration, Writing - original draft; Krzysztof J Gorgolewski, Conceptualization, Funding acquisition, Methodology, Project administration, Supervision, Writing – review and editing; Franklin Feingold, Project administration, Supervision, Writing – review and editing; Ross Blair, Nell Hardcastle, Oscar Esteban, Mathias Goncavles, Software, Writing – review and editing; Yaroslav O Halchenko, Conceptualization, Software, Writing – review and editing; Eric Miller, Conceptualization, Project administration, Software, Writing – review and editing; Joe Wexler, Data curation, Writing – review and editing; Anita Jwa, Investigation, Writing – review and editing; Russell Poldrack, Conceptualization, Formal analysis, Funding acquisition, Methodology, Project administration, Software, Supervision, Writing - original draft, Writing – review and editing

### Author ORCIDs

Christopher J Markiewicz ⓘ http://orcid.org/0000-0002-6533-164X
Krzysztof J Gorgolewski ⓘ http://orcid.org/0000-0003-3321-7583
Nell Hardcastle ⓘ http://orcid.org/0000-0002-3837-0707
Oscar Esteban ⓘ http://orcid.org/0000-0001-8435-6191
Russell Poldrack ⓘ http://orcid.org/0000-0001-6755-0259

### Decision letter and Author response

Decision letter https://doi.org/10.7554/eLife.71774.sa1
Author response https://doi.org/10.7554/eLife.71774.sa2

## Additional files

### Supplementary files
• Transparent reporting form

## Data availability

The OpenNeuro data repository is accessible at http://openneuro.org. The derived data used to generate the analyses and figures reported here are available at https://doi.org/10.5281/zenodo. 5559041.

The following dataset was generated:

| Author(s) | Year | Dataset title | Dataset URL | Database and Identifier |
|---|---|---|---|---|
| Poldrack R, Esteban O | 2021 | poldrack/OpenNeuro_ analyses: Revision and updates | https://doi.org/10. 5281/zenodo.5559041 | Zenodo, 10.5281/ zenodo.5559041 |

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
