## [Decision Letter]

**Acceptance summary:**

This manuscript provides a description of OpenNeuro, a data-sharing platform that is built upon the Brain Imaging Data Structure (BIDS) format. More than 600 data sets are currently stored in BIDS, following the FAIR principles, and integrated with data analysis tools. This is a highly important resource for the neuroimaging community, and the shared data sets have already been used in basic neuroscience research and for methods development.

**Decision letter after peer review:**

Thank you for submitting your article "OpenNeuro: An open resource for sharing of neuroimaging data" for consideration by *eLife*. Your article has been reviewed by 3 peer reviewers, and the evaluation has been overseen by a Reviewing Editor and Chris Baker as the Senior Editor. The following individuals involved in review of your submission have agreed to reveal their identity: Nico Dosenbach (Reviewer #1); Michael J Hawrylycz (Reviewer #2); Karel Svoboda (Reviewer #3).

All reviewers agreed that OpenNeuro is a valuable resource for the neuroimaging community and that your paper describing this platform is impeccable in scholarship, writing, visualization, and argumentation. Reviewers did not identify any essential issues but made a number of suggestions for how to further improve the paper. Please consider these suggestions (at your discretion) when preparing a revised manuscript.

*Reviewer #1 (Recommendations for the authors):*

This article is most important, timely, flawlessly argued and written. Everyone in neuroimaging owes a large debt to the creators of BIDS and OpenNeuro. The manuscript is very diplomatic. My only suggestions for further improvements would be to consider more explicitly discussing the rates of data sharing, data sharing willingness, amongst researchers and ways to encourage more of them to zealously share their data. For example, should more funding agencies across the globe require data sharing through an open, easy to use portal as a condition of funding. Should all scientific journals require it, de facto banning the 'available upon request' unsharing approach. Should they all require use of the same repository or is variety amongst platforms a strength? Along similar lines, could it further strengthen the manuscript to even more explicitly state that the PIs sharing their data are reaping large, direct benefits in terms of publications, citations, impact of their work and that refusal to share via an open platform is actually a suboptimal strategy. Any advice for researchers working in countries that make data sharing more difficult? Would it be too undiplomatic to suggest that the very largest datasets, specifically HCP, ABCD, UKB should also be added to OpenNeuro? Why not put all the neuroimaging data being collected in one place? Would there be any utility to adding a promotional angle to openneuro that posts/tweets data set download stats and boosts re-use papers.

*Reviewer #2 (Recommendations for the authors):*

1. The workflow schematic for data submission is clear and well organized, for an announcement of a resource such as this the authors might consider enhancing the image somewhat with icons of various steps. While clearly not essential, this may have the effect if attractively done in drawing more users to OpenNeuro.

2. I like the multiple dimensions of big data in terms of depth, width, breadth. These terms would do well to be used in experimental data practices beyond neuroimaging as well.

3. The resource considering the volume of data produced in the field is still quite small, and I think too much space in the paper may being used on describing the present datasets. This description does not necessarily have lasting value, as the deposited data may still be too small to be reflective of longer-term trends. I would considerably shorten this section, and for example, the word clouds might be eliminated.

4. Further, the section on present usage would fall under the same considerations, and might be shortened. Unless, for example, the authors a specialized vision for the types of neuroimaging datas they expect to be stored in OpenNeuro.

*Reviewer #3 (Recommendations for the authors):*

'discovery, reuse, repeatability, and reproducibility,' – repeatability vs reproducibility?

Define 'bespoke organizational scheme'.

---

## [Author Response]

Reviewer #1 (Recommendations for the authors):This article is most important, timely, flawlessly argued and written. Everyone in neuroimaging owes a large debt to the creators of BIDS and OpenNeuro. The manuscript is very diplomatic. My only suggestions for further improvements would be to consider more explicitly discussing the rates of data sharing, data sharing willingness, amongst researchers and ways to encourage more of them to zealously share their data. For example, should more funding agencies across the globe require data sharing through an open, easy to use portal as a condition of funding. Should all scientific journals require it, de facto banning the 'available upon request' unsharing approach.

While we agree in principle with the reviewer that funding agencies and journals should require data sharing in principle, we are reluctant to use this paper to make such a strong claim, as we worry that it might overshadow our discussion of the repository itself.

Should they all require use of the same repository or is variety amongst platforms a strength?

We believe that there is utility in a variety of platforms for data sharing. Most importantly, some datasets will not be shareable via the fully open CC0 model of OpenNeuro (for ethical or regulatory reasons), so platforms that support more restrictive data sharing will remain necessary. We have added a comment to this effect in the revision (p. 14).

Along similar lines, could it further strengthen the manuscript to even more explicitly state that the PIs sharing their data are reaping large, direct benefits in terms of publications, citations, impact of their work and that refusal to share via an open platform is actually a suboptimal strategy.

While there is some work suggesting that this is the case in general (Colavizza et al., 2020), we do not have data demonstrating this directly for data sharing via OpenNeuro, and we are thus reluctant to make strong claims about the benefits for OpenNeuro users in terms of the impact of their work.

Any advice for researchers working in countries that make data sharing more difficult?

We thank the reviewer for this open-ended suggestion. We have extended our discussion of Open Brain Consent resource (Bannier et al., 2021) (end of page 5) with a note in reference to EU’s GDPR, one of the most restrictive regulations affecting a substantial number of countries.

Would it be too undiplomatic to suggest that the very largest datasets, specifically HCP, ABCD, UKB should also be added to OpenNeuro? Why not put all the neuroimaging data being collected in one place?

Although we agree with the reviewer that centralization of all sharing of neuroimaging might be an appealing idea on a first thought, in practice there is a high risk that such a strategy would actually reduce the amount of data being shared. As noted above and on Page 14 in the revision, the nonuniformity of data sharing policies and regulations, as well as a range of other factors that determine how data may be shared, necessitates the existence of a variety of data sharing platforms. OpenNeuro is designed and promoted to cover the open and permissive end of the spectrum.

Would there be any utility to adding a promotional angle to openneuro that posts/tweets data set download stats and boosts re-use papers.

We acknowledge the utility of this suggestion, and have edited the manuscript on page 15 to indicate that we also devote OpenNeuro resources towards the neuroimaging community with blog posts and social media presence. We agree that promoting re-use papers would be useful, and we do this when possible; however, there is currently no effective automated way to discover reuse, so it requires substantial manual effort which is not sustainable in the long term.

Reviewer #2 (Recommendations for the authors):1. The workflow schematic for data submission is clear and well organized, for an announcement of a resource such as this the authors might consider enhancing the image somewhat with icons of various steps. While clearly not essential, this may have the effect if attractively done in drawing more users to OpenNeuro.

We have updated Figure 1 to be more attractive, with icons for each step.

2. I like the multiple dimensions of big data in terms of depth, width, breadth. These terms would do well to be used in experimental data practices beyond neuroimaging as well.

We appreciate this comment, and hope that the current manuscript can help spread these terms.

3. The resource considering the volume of data produced in the field is still quite small, and I think too much space in the paper may being used on describing the present datasets. This description does not necessarily have lasting value, as the deposited data may still be too small to be reflective of longer-term trends. I would considerably shorten this section, and for example, the word clouds might be eliminated.

We appreciate that the current data may not be reflective of longer-term trends, but we think that it is essential for this paper to outline the current contents of the archive in order to make clear the kinds of data that are being shared at present. We have, however, moved the word clouds to a supplementary figure in order to streamline the paper while still keeping the results available to interested readers and allowing them to be presented in a larger format.

4. Further, the section on present usage would fall under the same considerations, and might be shortened. Unless, for example, the authors a specialized vision for the types of neuroimaging datas they expect to be stored in OpenNeuro.

As with the previous comment, we think that an explanation of the present usage is essential for readers to understand the ways in which the database can be useful at present.

Reviewer #3 (Recommendations for the authors):'discovery, reuse, repeatability, and reproducibility,' – repeatability vs reproducibility?

We have streamlined this by removing “repeatability”.

Define 'bespoke organizational scheme'.

We have reworded this as “custom organizational scheme” to make it clearer.